# LC-AMP-F1 Derived from the Venom of the Wolf Spider *Lycosa coelestis*, Exhibits Antimicrobial and Antibiofilm Activities

**DOI:** 10.3390/pharmaceutics16010129

**Published:** 2024-01-19

**Authors:** Yuxin Song, Junyao Wang, Xi Liu, Shengwei Yu, Xing Tang, Huaxin Tan

**Affiliations:** 1Institute of Biochemistry and Molecular Biology, Hengyang Medical College, University of South China, Hengyang 421001, China; 2Hunan Key Laboratory for Conservation and Utilization of Biological Resources in the Nanyue Mountainous Region, College of Life Sciences, Hengyang Normal University, Hengyang 421002, China

**Keywords:** antimicrobial peptide, *Lycosa coelestis*, LC-AMP-F1, antibacterial, antibiofilm

## Abstract

In recent years, there has been a growing interest in antimicrobial peptides as innovative antimicrobial agents for combating drug-resistant bacterial infections, particularly in the fields of biofilm control and eradication. In the present study, a novel cationic antimicrobial peptide, named LC-AMP-F1, was derived from the cDNA library of the *Lycosa coelestis* venom gland. The sequence, physicochemical properties and secondary structure of LC-AMP-F1 were predicted and studied. LC-AMP-F1 was tested for stability, cytotoxicity, drug resistance, antibacterial activity, and antibiofilm activity in vitro compared with melittin, a well-studied antimicrobial peptide. The findings indicated that LC-AMP-F1 exhibited inhibitory effects on the growth of various bacteria, including five strains of multidrug-resistant bacteria commonly found in clinical settings. Additionally, LC-AMP-F1 demonstrated effective inhibition of biofilm formation and disruption of mature biofilms. Furthermore, LC-AMP-F1 exhibited favorable stability, minimal hemolytic activity, and low toxicity towards different types of eukaryotic cells. Also, it was found that the combination of LC-AMP-F1 with conventional antibiotics exhibited either synergistic or additive therapeutic benefits. Concerning the antibacterial mechanism, scanning electron microscopy and SYTOX Green staining results showed that LC-AMP-F1 increased cell membrane permeability and swiftly disrupted bacterial cell membranes to exert its antibacterial effects. In summary, the findings and studies facilitated the development and clinical application of novel antimicrobial agents.

## 1. Introduction

Biofilms are the lifestyle of many prokaryotic organisms, where planktonic bacteria gather together to resist the adverse external environment and evade the body’s immune defense mechanism and the infringement of antibiotics. The previous studies have demonstrated that the resistance of bacteria to antibiotics after forming biofilms is 1000 times higher than that in the floating state, which also increases the persistence and recurrence of bacterial infection [1,2]. Biofilms can cause many diseases, such as periodontitis [3], otitis media [4,5], osteomyelitis [6,7]. Furthermore, it should be noted that biofilms have the capability to adhere to various materials, including catheters [8,9,10], prosthetic joints [7,11], contact lenses [12], mechanical artificial heart valves [13], and others. This attachment often leads to the development of infections in these materials. The prevalence of biofilm infection has a significant impact on human well-being and presents a substantial risk to individuals. Consequently, there is an immediate need to devise effective strategies to combat biofilm formation and its associated complications. Currently, there exist two prevalent approaches: the physical and mechanical method, which involves the utilization of ultrasonic tools for cleansing; and the pharmacological modification method, which employs antibacterial drugs or antibiotics to prevent the formation of biofilms.

Antimicrobial peptides (AMPs) are a class of peptides that possess the ability to effectively eliminate a wide range of bacteria, and they exist widely in animals, plants and microorganisms. In addition to their antibacterial properties, AMPs have antiviral, anti-tumor, anti-inflammatory, and synergistic effects when used in conjunction with antibiotics [14,15,16]. Numerous studies have demonstrated that AMPs can inhibit the formation of bacterial biofilms and eradicate mature biofilms [15,17]. Kang et al. [18] observed that the AMP LL-37 had a scavenging effect on the biofilm produced by *Staphylococcus aureus* (*S. aureus*). Colombo et al. [19] discovered that LL-37 could interact with the cell membrane of *Streptococcus mutans* (*S. mutans*) to prevent dental caries. Like LL-37, many other AMPs possess biofilm inhibitory activities, and they may be developed as potential drugs for preventing and treating various infectious diseases.

The peptide LC-AMP-F1 was identified from the venom gland cDNA library of spider *L. coelestis*, and the preliminary determination indicated that it had certain antibacterial activity. Coincidentally, LC-AMP-F1 exhibited the same sequence with that of LyeTx II, which was isolated from the venom of the spider *Lycosa erythrognatha* by Nunes research group [20]. They obtained LyeTx II sequence by Edman degradation and mass spectrometry (ESI-Q-TOF and MALDI-TOF-TOF), and performed biochemical analysis using the synthetic form of the peptide. In addition, they found that the peptide LyeTx II showed tumor-promoting activity in MDA-MD-231 breast cancer cells [21]. However, other biological activities of LC-AMP-F1 have not yet been reported in the literature. The aim of this study is to investigate the potential abilities of LC-AMP-F1 for antibacterial activity and cytotoxicity towards erythrocytes in vitro. Hence, in this work, melittin, a classical membrane-breaking AMP derived from honeybee venom, was set as a positive control [22,23,24]. The antimicrobial resistance of LC-AMP-F1 and melittin were analysed with bacterial strains from China General Microbiological Culture and clinic. Our results showed that LC-AMP-F1 had excellent antibacterial activity, temperature and acid-base stability, biofilm inhibition and eradication effects, and low toxicity.

## 2. Materials and Methods

### 2.1. Identification and Bioinformatics Analysis of LC-AMP-F1 Sequence

The *L. coelestis* venom glands were dissected, and their total RNA was extracted with Trizol reagent. The venom gland cDNA library was constructed following the instructions from the SMART cDNA Library Construction Kit. Colony PCR was performed and the PCR products were sequenced as described in previous work [25,26]. The cDNA sequence was translated into peptide sequences by the MEGA 7.0 software. The secondary structure of LC-AMP-F1 was predicted by I-TASSER (https://zhanggroup.org/I-TASSER/). The physiochemical characteristics of LC-AMP-F1 was assessed using ProtParam (https://web.expasy.org/protparam/), and the helical wheel projection of LC-AMP-F1 was analyzed by HeliQuest (https://heliquest.ipmc.cnrs.fr/cgi-bin/ComputParams.py). All analysis results were accessed on 24 July 2022.

### 2.2. Circular Dichroism (CD) Spectroscopy

LC-AMP-F1 was dissolved in water and a 50% (*v*/*v*) solution of trifluoroethanol (TFE) at final peptide concentration of 500 μg/mL and afterwards introduced into a quartz colorimeter equipped with a light diameter of 1 mm. The resulting solution was then subjected to analysis using a circular dichroic spectrometer, which operated within a wavelength range of 190–260 nm.

### 2.3. Bacterial Strains

The China General Microbiological Culture Collection Centre provided *Salmonella typhimurium* (*S. typhimurium*) CGMCC 1.1174, *Shigella dysenteriae* (*S. dysenteriae*) CGMCC 1.1869, and *Pseudomonas aeruginosa* (*P. aeruginosa*) CGMCC 1.596. The National Centre for Medical Culture Collections provided *S. aureus* CMCC 26003. We acquired MRSA (methicillin-resistant *S. aureus*) ATCC 43300 from the American Type Culture Collection.

From October 2021 to April 2023, all clinical isolates analysed in this study were acquired from the University of South China’s Affiliated Nanhua Hospital. Serum, urine, sputum, or blood from patients were used to isolate multidrug-resistant (MDR) strains of *Enterococcus faecium* (*E. faecium*), *S. aureus*, *Klebsiella pneumoniae *(*K. pneumoniae*), *Acinetobacter baumannii *(*A. baumannii*), *P. aeruginosa*, and *Enterobacter species *(*E. species*).

### 2.4. Determination of Minimum Inhibitory Concentration (MIC)

First, the standard (*E. coli*, *S. typhimurium*, *S. dysenteriae*, *P. aeruginosa*, *P. vulgaris*, *S. albus* and MRSA) and clinical (*E. faecium*, *S. aureus*, *K. pneumoniae*, *A. baumannii*, *P. aeruginosa*, and *E.* species) bacterial strains were activated and cultured overnight to the logarithmic growth phase for subsequent experiments. After the bacterial solution was diluted to 10^5^ CFU/mL, it was added to the 96-well plate. The AMP (containing experimental group LC-AMP-F1 or positive control group melittin) was mixed with the bacterial solution in the 96-well plate, and gradient dilution was performed by the micro-double dilution method. The light absorption value was measured at 600 nm after incubation for 16 h. The MIC with an inhibition rate greater than 95% was the MIC value. The inhibition rate is calculated as follows:Inhibition % = (A_negative_ − A_sample_)/A_negative_.

### 2.5. Stability Experiment

The study employed *E. coli* as the experimental subject, with melittin serving as the positive control.

Thermal stability: The AMPs were pre-heated in a water bath at 80 °C and 100 °C, respectively, for 1 h for MIC experiments.

Acid-base stability: First, 5 M HCl and 5 M NaOH were prepared and added to MH medium so that their pH values were 5, 6, 8, and 9 respectively. Then, the AMPs were added to MH medium of different pH for the MIC experiment.

Salt ion stability: A metal salt ion solution of 150 mM NaCl, 4.5 mM KCl, 6 μM NH_4_Cl, 1 mM MgCl_2_ and 2.5 mM CaCl_2_ was prepared in advance. After sterilization, 1/10 volume of metal ion salt solution was added to the bacterial solution to determine the change in its MIC value.

### 2.6. Time-Kill Kinetics

The suspension of *E. coli* was adjusted to a concentration of 10^5^ CFU/mL, and the diluted bacterial solution was mixed with different concentrations of peptides. The colonies were counted at 0, 15, 30, 60, 90 and 120 min, respectively, after incubation for 24 h. The positive control in the study was melittin, while the negative control was PBS.

### 2.7. Biofilm Inhibition and Eradication Assays

In this experiment, *E. coli* and clinical *S. aureus* 1065 were utilized as the experimental subject, whereas melittin was employed as the positive control.

Biofilm inhibition: Bacterial solution of 100 μL diluted to 10^5^ CFU/mL and AMPs of 100 μL with different concentrations were added to the 96-well plate, which was incubated for 24 h and washed 3 times with PBS, then fixed with methanol for 15 min, stained with 0.1% crystal violet for 10 min, washed with water, and finally dissolved with 33% ice acetic acid. The absorption value at 595 nm was determined.

Biofilm eradication: The diluted bacterial solution was incubated for 24 h and AMPs of different concentrations were added. After incubating for 24 h, PBS was used for washing, and the subsequent steps such as fixed staining were the same as the biofilm inhibition experiment.

### 2.8. Antibiotic Synergy

The determination of the synergistic effect of LC-AMP-F1 and melittin in combination with antibiotics was conducted using the checkerboard method. In 96-well plates, 90 μL of *E. coli* suspension diluted to 10^5^ CFU/mL with MH medium was added to each well. The highest concentration of AMPs and antibiotics (erythromycin and levofloxacin) started at 2 × MIC concentration. The concentration of AMP in each row was decreased by 1/2, as same as the concentration of antibiotics in each column. The formula for calculating the fractional inhibitory concentration index (FICI) is as follows:FICI = (MIC_A’_/MIC_A_) + (MIC_B’_/MIC_B_)

MIC_A_ and MIC_B_ represent their respective MIC when A and B are used alone.

MIC_A’_ and MIC_B’_ represent the MIC when A and B are combined, respectively.

FICI ≤ 0.5 is the synergistic effect. A total of 0.5 < FICI ≤ 1 is partial synergy (additive effect); If 1 < FICI ≤ 2, it is indifferent. FICI > 2 indicates antagonism.

### 2.9. SYTOX Green Staining Assay

The *E. coli* cells at logarithmic growth stage were harvested after centrifuging at 4000 rpm for 3 min, then washed with 10 mM Tris-HCl 3 times. After suspension, the concentration of bacterial solution was adjusted to 10^8^ CFU/mL. SYTOX green dye was pre-incubated with bacterial solution for 10 min and AMPs of different concentrations were added. The fluorescence intensity per minute within 1 h was measured, and the parameters were set as: excitation light 485 nm, emission light 525 nm, gain 50. The positive control of the experiment involved the utilization of melittin, while the negative control involved the utilization of BSA.

### 2.10. Scanning Electron Microscope (SEM)

The LC-AMP-F1 and melittin (positive control group) with a final concentration of 4 × MIC were added to the suspension of *E. coli*, diluted to 10^8^ CFU/mL, incubated at 37 °C for a certain time, and then centrifuged at 4000 rpm for 10 min. The precipitate was washed three times with PBS and then fixed with 4% paraformaldehyde at 4 °C for 12 h. After being washed three times with PBS, gradient dehydration of each sample was carried out with 20%, 50%, 80% and 100% ethanol for 10 min each time, and finally re-suspended with 100% ethanol. The samples were transferred and dried on silicon wafers at 60 °C for 5 min. Then, specimens are attached to metallic stubs using carbon stickers and sputter-coated with gold for 30 s by the Lon Sputtering Apparatus (HITACH MC1000, Tokyo, Japan). At last, the samples were observed by SEM (HITACHI Regulus 8100, 1 kV, Tokyo, Japan) [27,28].

### 2.11. Cytotoxicity Assay

The normal cells (LO2 and HEK293T) and cancer cells (4T1) at logarithmic phase were diluted with DMEM medium to 10^4^ cells per well and incubated in the cell incubator (37 °C, 5% CO_2_) for 12 h to make the cells adhere to the wall. Then, different concentrations of LC-AMP-F1 and melittin were added. After 12 h of treatment, 10 μL of CCK-8 was added to each well, and the absorption value at 450 nm was measured after incubation for 0.5–4 h. The cell survival rate was calculated as:Survival rate (%) = (OD_sample_ − OD_blank_)/OD_negative_

### 2.12. Hemolysis Assay

Fresh rabbit blood was collected and centrifuged at 4 °C at 1200 rpm for 5 min to obtain erythrocyte precipitation. Then, the cells were washed with 1 × PBS 2–3 times until the supernatant was nearly transparent. Subsequently, 1% (*v*/*v*) erythrocytes were prepared with 1 × PBS resuspension. Different concentrations (5–320 μM) of LC-AMP-F1 and melittin were added and incubated at 37 °C for 1 h. Finally, the OD_450_ of the supernatant of each sample was measured.

## 3. Results

### 3.1. Identification and Bioinformatics Analysis of LC-AMP-F1 Sequence

Based on the venom gland cDNA library of *L. coelestis*, we identified a new AMP gene which encoded the peptide (named LC-AMP-F1) (Figure 1). The peptide LC-AMP-F1 composes of 19 amino acid residues and its sequence is AGLGKIGALIQKVIAKYKA-NH_2_ (Figure 1a). Surprisingly, LC-AMP-F1 has identical peptide sequence with that of LyeTx II [20]. LC-AMP-F1 with a calculated molecular mass of 1942.42 contains four positively charged residues (Lys). Its theoretical isoelectric point and average of hydropathicity (GRAVY) are 10.48 and 0.574, respectively.

I-TASSER was used to simulate the three-dimensional structure of LC-AMP-F1, and the results demonstrated that its significant portion of the structure exhibited alpha-helical conformation (Figure 1b). The distribution of hydrophobic and hydrophilic residues of LC-AMP-F1 sequence was displayed in an α-helical wheel, as shown in Figure 1c, predicted by Heliquest. CD spectroscopy can be successfully used to study the secondary structure of peptides in two different environments (water and 50% TFE), and the TFE/water mixtures were applied to stabilize secondary-structure formation in peptides [29,30]. As can be seen from Figure 1d, there is a positive peak at 192 nm and a negative peak at 208 nm and 222 nm, respectively, indicating that LC-AMP-F1 presents an α-helical conformation in hydrophobic environment. However, it adopts a random coil conformation in aqueous environments.

### 3.2. Antibacterial Activities of LC-AMP-F1 In Vitro

To validate its potential biological activity, we systematically evaluated the antimicrobial activity of LC-AMP-F1 using a variety of standard microbial and clinical isolates (Figure 2). As shown in Figure 2a, LC-AMP-F1 strongly inhibited three gram-negative bacterial strains, including *E. coli*, *S. typhimurium*, and *S. dysenteriae*, with MICs ranging from 5 to 10 μM. Whereas, it had a weak or negligible effect on two gram-negative bacteria (*P. aeruginosa* and *P. vulgaris*) and two gram-positive strains (*S. aureus* and MRSA). However, the inhibitory activity of LC-AMP-F1 against the five clinical isolates was highly significant and comparable to that of melittin. The MICs of LC-AMP-F1 for the five clinical isolates remained within 10 μM, as illustrated in Figure 2b. Particularly, the MICs for *E. faecium* and *P. aeruginosa* were merely 2.5 μM. Notably, all of these clinical isolates exhibited multi-drug resistance and are the major pathogens in nosocomial infections. This suggests that LC-AMP-F1 exhibits bacteriostatic effects against a wide range of microorganisms and may have different mechanisms of action from those of conventional antibiotics.

To assess the stability of LC-AMP-F1, a series of experiments were conducted to investigate its temperature, acid-base, and salt ion stability. The results in Appendix A showed that LC-AMP-F1 had favorable stability under high temperature and weak base (pH = 8) treatment, and its antimicrobial efficacy was slightly decreased under weak acid environment. Similar to melittin, the inhibitory activity of LC-AMP-F1 declined in the rising ionic environment, but its MIC against *E. coli* stayed within the effective concentration range at 20 μM (Appendix A). Interference with the electrostatic interactions between the AMP and the cell membrane may be the cause of the effect of the increased ions on the peptide. The two cationic AMPs, for instance, were more strongly affected by the more positively charged divalent metal ions. This finding implied that the divalent cations competitively bind the negatively charged bacterial cell membranes, increasing the effective concentration of AMPs.

The killing kinetics of different concentrations of LC-AMP-F1 on *E. coli* were measured and shown in Figure 3. LC-AMP-F1 had an obvious rapid killing effect with a concentration of 8 × MIC at 15 min or a concentration of 4 × MIC at 30 min, while *E. coli* was almost killed at both concentrations at 120 min.

### 3.3. Biofilm Inhibition and Eradication Activities of LC-AMP-F1

Biofilms are an important cause of chronic and recurrent bacterial infections in the clinical practice, as well as an important factor contributing to the failure of conventional antibiotic therapies [31]. Here, we validated the biofilm inhibition and eradication ability of LC-AMP-F1 using the two biofilm structures formed by *E. coli*, planktonic and adherent (Figure 4). As illustrated in Figure 4a, the growth of *E. coli* biofilms were efficaciously inhibited by LC-AMP-F1 at a lower concentration, as the inhibition rate reached 80% at 5 μM. The inhibitory influence of LC-AMP-F1 on *E. coli* biofilm was found to outperform melittin in the concentration range of 0.3 to 20 μM, with a concentration-dependent rise. However, LC-AMP-F1 exhibited a restricted capacity to eliminate *E. coli* biofilm, exceeding melittin at a working concentration of ≤2.5 μM and achieving a maximal eradication rate of approximately 50% at a working concentration of 20 μM (Figure 4b). On the other hand, melittin was still able to extend its biofilm scavenging effect with increasing concentration and reached a maximum clearance (around 80%) at a concentration of 10 mM. Meanwhile, we also conducted experiments on clinical *S. aureus* 1065 biofilm, and the results are shown in Appendix A. The inhibition and eradication ability of LC-AMP-F1 on 1065 biofilm were weaker than those of melittin.

### 3.4. Antibiotic Synergy of LC-AMP-F1

Given the superior efficacy of LC-AMP-F1 in combating biofilms compared to melittin, particularly at doses below 2.5 μM. Hence, we conducted an investigation to assess the synergistic effect of LC-AMP-F1 in conjunction with two antibiotics, employing a micro-checkerboard dilution method. Briefly, the MICs of peptides and antibiotics were remeasured when they were used in combination at a series of fixed sub-MIC concentrations (peptides or antibiotics alone). The MICs in combination were recorded, and the FICIs were calculated in Figure 5. A synergistic inhibitory effect of erythromycin and LC-AMP-F1 was observed on the growth of the tested *E. coli* cells, resulting in a complete halt to bacterial proliferation. This effect (FICI = 0.5) was achieved with a peptide and antibiotic concentration of merely one-fourth of the initial MICs. In conjunction with levofloxacin, the additive effect of LC-AMP-F1 was the effective dose of both tested agents by half. The FICI data indicate that the synergistic effect of LC-AMP-F1 with both antibiotics was more significant in comparison to melittin, as indicated by the lower FICI values.

### 3.5. Antibacterial Mechanism of LC-AMP-F1

Considering its amphiphilic α-helix secondary structure, the antimicrobial activity of LC-AMP-F1 and its synergistic effect with antibiotics are likely to be related to the membrane-interacting ability of the peptide. To verify this hypothesis, we explored the impact of the peptide on the permeability of bacterial cell membranes by incubating peptide-treated *E. coli* with SYTOX Green, a DNA-binding dye that is unable to permeate intact cell membranes [32]. A concentration-dependent characteristic was observed in the fluorescence intensity of bacterial suspensions stained with SYTOX, as illustrated in Figure 6. The fluorescence intensity of the increase was substantial as the peptide action time was prolonged. This process happened as a result of SYTOX entering the cytoplasm due to cytoplasmic membrane permeabilization. Once across the cytoplasm, SYTOX binds to nucleic acids and significantly increases fluorescence emission. Compared to melittin, the fluorescence intensity of LC-AMP-F1-treated bacteria exhibited a reduced increase. However, the interaction between both peptides led to the quick binding of intracellular DNA to the dye within a time frame of less than 10 min. This observation aligns with our previous findings that demonstrated the rapid bactericidal efficacy of LC-AMP-F1. It is worth mentioning that LC-AMP-F1 had limited efficacy in inhibiting bacteria at concentrations below the MIC of 5 μM. However, it had a certain degree of influence on the permeability of bacterial cell membranes, which could probably account for the observed synergistic effect of LC-AMP-F1 at sub-MIC concentrations.

In order to further investigate the alterations in the morphology of the bacterial cell surface subsequent to the administration of LC-AMP-F1, SEM was employed. As illustrated in Figure 7, we were able to discern the influence of the peptide on the *E. coli* cell membrane. Ten minutes after being incubated with LC-AMP-F1 at a concentration four times the MIC, the cell surface of *E. coli* cells ruptured and collapsed visibly. In the meanwhile, as the duration of peptide treatment increased, the severity of disruption to the cell membrane progressively aggravated. Specifically, after 60 min, a considerable quantity of cellular structures experienced extensive disruption, resulting in a release of cellular components and the adhesion of cellular debris.

### 3.6. The Cytotoxicity of LC-AMP-F1

In order to evaluate the biosafety of the peptide, the cytotoxic activities of LC-AMP-F1 against eukaryotic cells were evaluated using three cell lines and erythrocytes (Figure 8). Evidently, its hemolysis was undetectable at 160 μM, as illustrated in Figure 8a. In contrast, the hemolytic activity of melittin was significantly higher than that of LC-AMP-F1, rupturing 98% of the erythrocytes at a final concentration of 5 μM. Furthermore, LC-AMP-F1 maintained a minimal level of toxicity towards the remaining three eukaryotic cell lines (Figure 8b–d). The growth of the assessed eukaryotic cells was unaffected at an action concentration of 20 μM. However, the toxicity of melittin became evident at a final concentration of 5 μM for 4T1 and HEK293T cells (inhibition rates of 59.4% and 65.3%, respectively). The findings indicate that LC-AMP-F1 exhibits a superior biosafety profile and demonstrates selectivity against bacterial infections at relatively high concentrations, in contrast to the generalized cytotoxic activity of melittin.

## 4. Discussion

Biofilm infections provide substantial risks as a result of their resistance to drugs, tendency to recur, and ability to become chronic infections [33]. These microorganisms exhibit a significant degree of resistance to antimicrobial agents, necessitating the administration of elevated dosages or the development of novel therapeutic approaches. Recurrences of biofilms are frequently observed due to their ability to re-establish following therapy [34,35]. Chronic infections have the potential to result in significant health consequences and hinder the efficacy of standard treatment methods. In the present study, the transcriptomic technique was employed to investigate the venom peptides of *L. coelestis*, and the main objective was to find novel antimicrobial peptides, especially active peptides combating drug-resistant bacteria and biofilms. According to the bioinformatics study, LC-AMP-F1 containing 19 amino acid residues was expected to possess an amphiphilic property characterized by an α-helical structure. To assess the potential utility of LC-AMP-F1 in the field of drug discovery, a comprehensive set of in vitro tests was undertaken to elucidate its antibacterial properties and underlying mechanisms. LC-AMP-F1 exhibits significant prospects for potential application as a therapeutic or tool agent with antimicrobial and antibiofilm abilities, as evidenced by thorough systematic investigation and comparison. Furthermore, its druggability has been further confirmed.

Initially, it was shown that LC-AMP-F1 displayed a wide range of antibacterial activity, effectively impeding the growth of various bacteria, including multidrug-resistant strains commonly associated with nosocomial infections. Also, it hindered the formation of biofilms. Moreover, LC-AMP-F1 promptly interacted with the bacterial cell membranes, leading to the disruption of their structural integrity and permeability, leading to cell lysis and subsequent bacterial death. Additionally, the LC-AMP-F1 exhibited a relatively simple spatial conformation, characterized by a single α-helical secondary structure. This particular configuration might contribute to its inherent stability, enabling it to endure extreme conditions such as elevated temperatures, acidic and alkaline environments, and fluctuations in ionic strength [28,36,37,38]. These attribute sets distinguished LC-AMP-F1 from most bioactive molecules, providing it with a distinct advantage. Furthermore, the capacity of LC-AMP-F1 to act on membranes in a concentration-dependent manner has significant prospects for its utilization in conjunction with other pharmaceutical agents as well as its application as a molecular tool for drug delivery. Finally, in contrast to the majority of cationic peptides that could penetrate membranes, LC-AMP-F1 exhibited minimal hemolytic activity and did not display any harmful effects on the eukaryotic cells that were tested. This finding highlighted the notable biosafety of LC-AMP-F1, as it exhibited selective activity against bacterial cells across a broad concentration range without inducing additional adverse reactions. However, it also implied that the antibacterial and antibiofilm properties of LC-AMP-F1 were not solely attributed to its amphiphilic membrane-interacting structure but rather resulted from a complicated balance between its toxicity and safety.

It is worth mentioning that many antimicrobial peptides from venomous animals [39], such as melittin [40], have antitumor effects. But surprisingly, the previous study showed that LyeTx II from *L. erythrognatha* venom increased MDA-MB-231 aggressive breast cancer cell proliferation by upregulating activation of p38 and NF-κB pathways and had non-toxicity to three other different breast cancer cell lines (MCF-7, MACL-1 and MGSO-3) [20,21]. In this work, LC-AMP-F1 was identified from *L. coelestis* venom and had an identical sequence with that of LyeTx II. The result indicated that LC-AMP-F1 had low toxicity to 4T1 murine breast cancer cells, and its negligible toxicity to breast cancer cells was consistent with the experimental results of LyeTx II. On the other hand, melittin suppressed the HIF-1α signaling pathway in the breast adenocarcinoma cell line MDA-MB-231 through downregulation of NF-κB gene expression [41] and inhibited the EGF-induced MMP-9 expression via blocking the NF-κB and PI3K/Akt/mTOR pathways in two breast cancer cells, MDA-MB-231 and MCF-7 cells [42]. Therefore, LC-AMP-F1 (or LyeTx II) and melittin have opposite regulatory effects on the NF-κB pathway.

The emergence of bacterial resistance and the creation of biofilm are outcomes that arise from the interplay of diverse biological mechanisms, including gene expression regulation, cellular communication, and immune response, among others [43]. Despite the apparent simplicity of the LC-AMP-F1 structure, various biological effects observed at varying concentrations underscores the complexity of its action mechanism. From another perspective, the LC-AMP-F1 sequence’s comparatively uncomplicated structure can serve as a molecular template for investigating the conformational correlation of antimicrobial peptides. Further investigation into molecular mechanisms of membrane action for peptide activity with the antibacterial and antibiofilm is the intended trajectory of our research, in addition to the pursuit of generating more active and selective drug molecules through the modification or reformulation of LC-AMP-F1.For example, we will consider how to eliminate tumor-promoting activity and enhance antibacterial activity based on LC-AMP-F1 as a template for molecular design. 

## Figures and Tables

**Figure 1 pharmaceutics-16-00129-f001:**
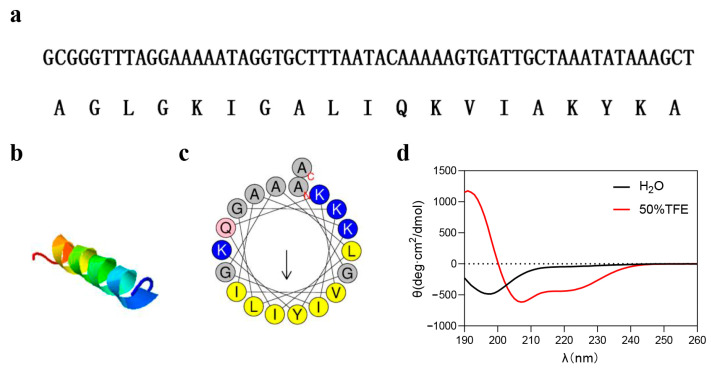
Structure of LC-AMP-F1. (**a**) The cDNA sequence and amino acid sequence of LC-AMP-F1. (**b**) Three-dimensional model from I-TASSER simulation of the secondary structure of LC-AMP-F1. (**c**) The α-helical wheel projection of LC-AMP-F1 presented by HeliQuest. (**d**) CD spectra of LC-AMP-F1 in water and 50% TFE solution. The hydrophobic residues are yellow, positively charged hydrophilic residues are blue, and negatively changed hydrophilic residues are pink. The N terminal is A, and the C terminal is A. The arrow indicates that it begins at 12 o ’clock and reads clockwise down the line.

**Figure 2 pharmaceutics-16-00129-f002:**
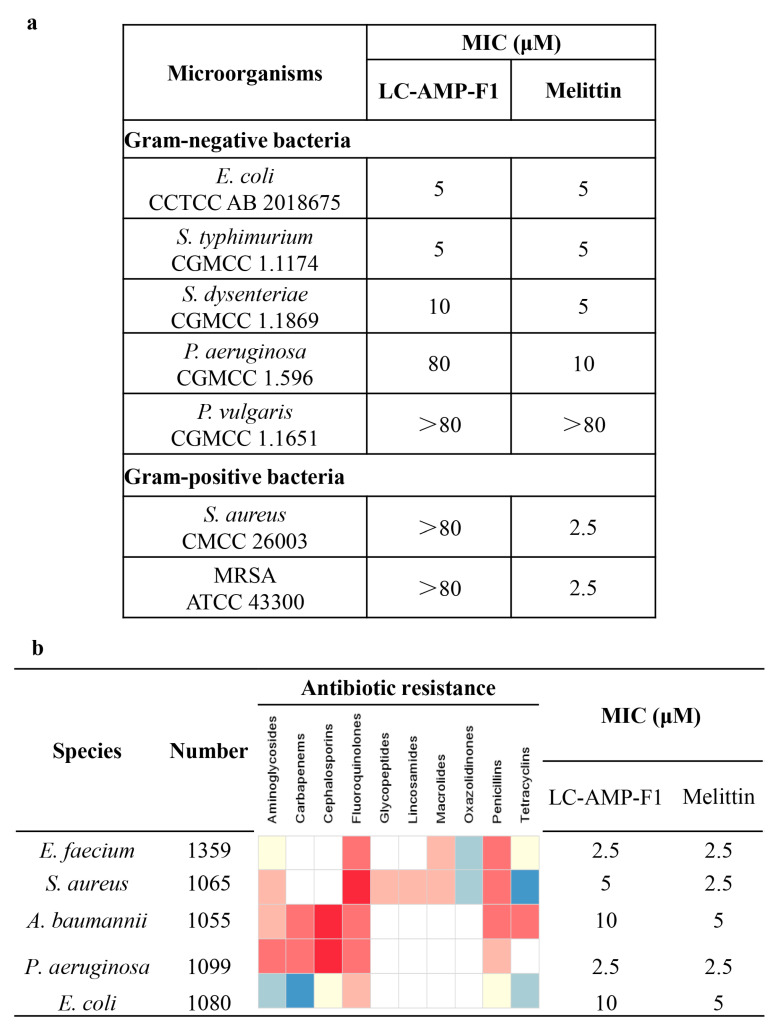
Antibacterial activity of LC-AMP-F1. MIC values of LC-AMP-F1 against standard strains (**a**) and clinical bacterial strains (**b**). The depth of the color indicates the level of resistance or sensitivity of the bacteria. The darker the color display, the higher the resistance/sensitivity to the antibiotic indicate. Blank boxes are shown if the susceptibility to agents in that class is not assessed.

**Figure 3 pharmaceutics-16-00129-f003:**
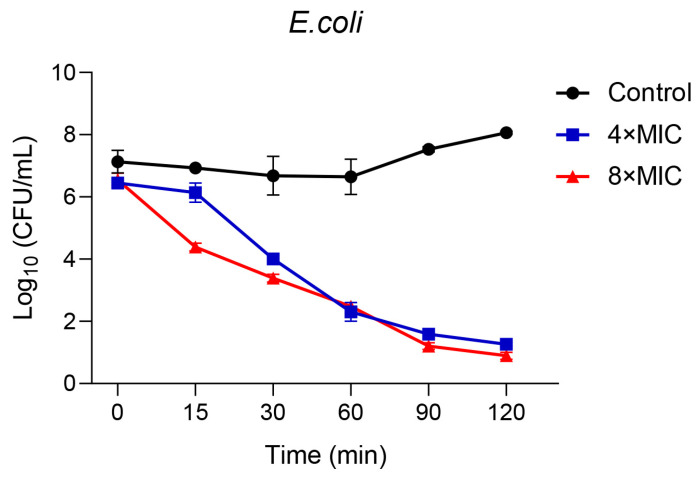
Time-killing curve of LC-AMP-F1 against *E. coli*. Zero min represents that the bacterial suspension is counted immediately after adding the sample. The other samples were collected at 15, 30, 60, 90, and 120 min.

**Figure 4 pharmaceutics-16-00129-f004:**
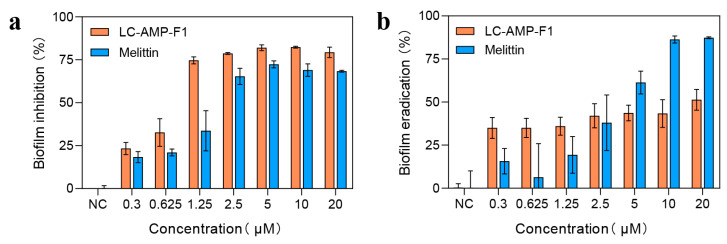
Inhibition (**a**) and eradication (**b**) activities of LC-AMP-F1 on *E. coli* biofilm.

**Figure 5 pharmaceutics-16-00129-f005:**
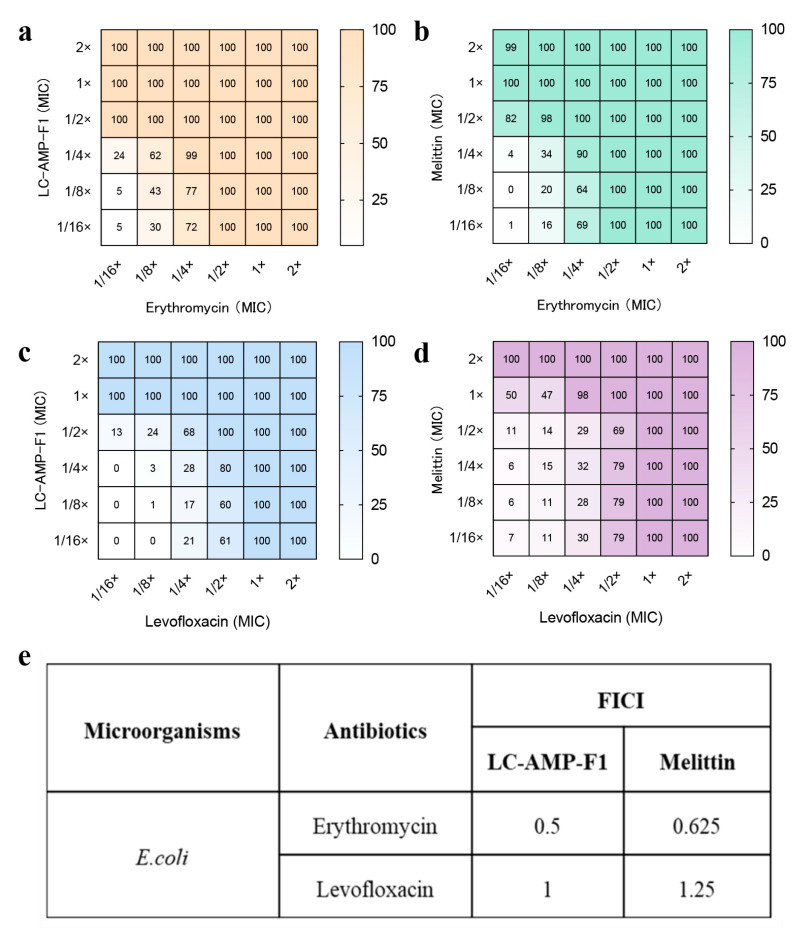
Checkerboard experiment in *E. coli*. The concentrations of LC-AMP-F1 and melittin in combination with erythromycin (**a**,**b**) and levofloxacin (**c**,**d**) were determined, respectively. (**e**) FICI of LC-AMP-F1 and melittin.

**Figure 6 pharmaceutics-16-00129-f006:**
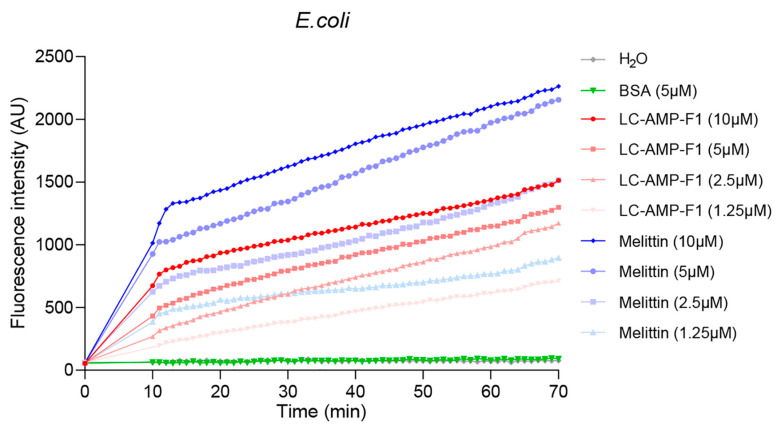
Changes of bacterial membrane permeability treated with LC-AMP-F1. The fluorescence intensity per minute was measured by SYTOX Green. Melittin was the positive control and BSA was the negative control.

**Figure 7 pharmaceutics-16-00129-f007:**
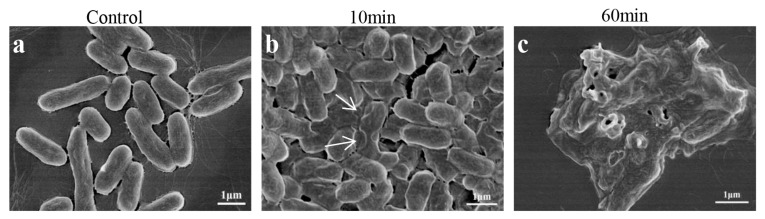
SEM images of *E. coli* treated with LC-AMP-F1. (**a**–**c**) represent negative control, treatment for 10 min and 60 min, respectively. The arrows indicate the typical damage to the plasma membranes of the bacteria.

**Figure 8 pharmaceutics-16-00129-f008:**
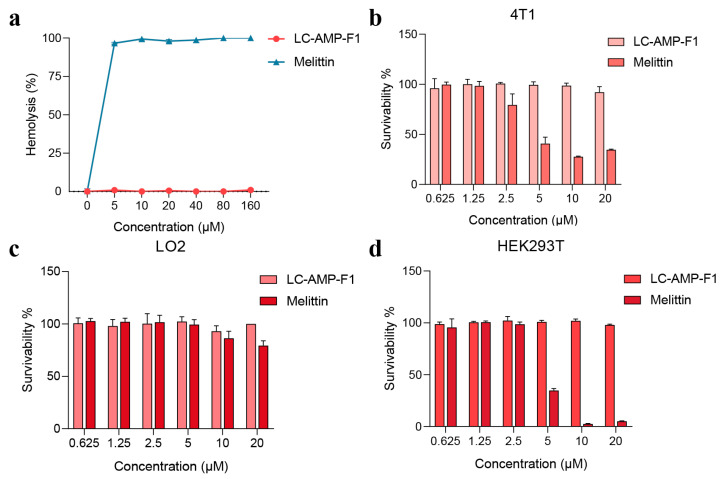
Hemolytic activity and cytotoxicity of LC-AMP-F1. (**a**) Hemolysis of LC-AMP-F1 against rabbit erythrocytes. Toxicity of LC-AMP-F1 to cancer (**b**) and normal (**c**,**d**) cells.

## Data Availability

The data presented in this study are available on request from the corresponding author.

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
