# Peer review of "LC-AMP-F1 Derived from the Venom of the Wolf Spider Lycosa coelestis, Exhibits Antimicrobial and Antibiofilm Activities"

_pharmaceutics, 2024, doi:10.3390/pharmaceutics16010129_

Round 1
Reviewer 1 Report
Comments and Suggestions for Authors
Please consider addressing the following issues
- It is not clear how exactly the peptide was obtained (eg. Synthesis, isolation from the venom). Please specify and provide source. This aspect is only mentioned for the LyeTx II peptide in line 65.
- Considering the antimicrobial and antibiofilm determinations: it would have been preferable to also add as standards some antibiotics that are approved medicines (this is widespread practice even in the article that the authors cite as references).
- Considering the antibiofilm activity: why did the authors only test against E coli biofilm production? Consider a more broad-spectrum testing.
- In regards of article originality: it is not clear whether the active peptide from this study LC-AMP-f1 is in fact identical with the LyeTx II. Line 192 suggests that they are, while line 63 hints they are just similar. In this case the relevance of the present research needs to be reiterated.
- Check how some articles have been cited.
Author Response
- It is not clear how exactly the peptide was obtained (eg. Synthesis, isolation from the venom). Please specify and provide source. This aspect is only mentioned for the LyeTx II peptide in line 65.
RESPONSE: Thanks very much for the comment. “L. coelestis venom glands were dissected, and their total RNA was extracted with Trizol reagent. The venom gland cDNA library was constructed following the instructions for the SMART cDNA Library Construction Kit. Colony PCR was performed and the PCR products were sequenced as described in previous work. ” have been added to “2. Materials and methods”. Other relevant content has been added and marked in red at the appropriate places in the latest manuscript.
- Considering the antimicrobial and antibiofilm determinations: it would have been preferable to also add as standards some antibiotics that are approved medicines (this is widespread practice even in the article that the authors cite as references).
RESPONSE: Thank you for your valuable suggestion. In fact, in our previous work, ampicillin sodium was used as an antibiotic control. However, the effective dose of antibiotics was greatly increased due to the problem of resistance of the clinical strains used. Indeed, in similar current anti-biofilm studies, antibiotics are usually used in high doses or in combination, which are not effectively comparable to the peptide molecules used in this paper. Therefore, in this paper, we used Mellitin, a mature membrane-breaking peptide that is structurally closer to LC-AMP-F1, as a control.
- Considering the antibiofilm activity: why did the authors only test against E.colibiofilm production? Consider a more broad-spectrum testing.
RESPONSE: Thank you for your valuable suggestions. We have supplemented the biofilm inhibition and eradication experiments of the peptide against clinical Staphylococcus aureus 1065 (Figure S1), and the relevant content has been supplemented in the corresponding position in the main text and marked in red font.
- In regards of article originality: it is not clear whether the active peptide from this study LC-AMP-F1 is in fact identical with the LyeTx II. Line 192 suggests that they are, while line 63 hints they are just similar. In this case the relevance of the present research needs to be reiterated.
RESPONSE: Thank you. The “consensus” in line 63 has been modified to the “same”.Although the primary sequence information of LC-AMP-F1 is consistent with that of LyeTx II, the two peptides do not have the same origin. On the other hand, the present work focuses on the antimicrobial potential of LC-AMP-F1, whereas studies on LyeTx II have mainly focused on antitumor aspects. Based on your suggestion, we have added relevant content in the Discussion section being labeled in red font.
- Check how some articles have been cited.
RESPONSE: Thanks for your comments. Based on your suggestions, we have scrutinized and adjusted the references.
Reviewer 2 Report
Comments and Suggestions for Authors
A very well written manuscript focused on a very important and actual subject.
Overall the manuscript presents the characterization and testing of peptides to access their antimicrobial and antibiofilm capacities.
The bacterial strain tested included both Gram(+) and Gram(-) as well as coccus and roads. By choosing these bacteria the authors cover the eventual influence of membrane cell composition and morphological shape on a potential response of the studied peptide.
As previously stated this is a very well thought and presented manuscript (one of the best from the last couple of years that I have reviewed), although I do have some comments/suggestions:
- - In all the text please write Lycosa coelestis in italic Lycosa coelestis.
- - The discussion needs to have more comparisons with what it is already published. In fact, throughout the discussion there are only 3 references (from 35 to 37) and in the first couple of sentences. The rest of the discussion is not supported by the published previous studies from other authors.
- - All information from lines 399 to 405 is missing and MUST be added
- - References: at least 23 of the 38 references are prior to the current decade (2020-2023). The authors must refresh this list with more updated references considering the amount of work published due to the importance of the subject. Moreover, some of the references are not complete (for example references 10 and 11).
Author Response
- In all the text please write Lycosa coelestis in italic Lycosa coelestis.
RESPONSE: Thank you for pointing this out. We have made the changes you suggested in the latest manuscript.
- The discussion needs to have more comparisons with what it is already published. In fact, throughout the discussion there are only 3 references (from 35 to 37) and in the first couple of sentences. The rest of the discussion is not supported by the published previous studies from other authors.
RESPONSE: Thanks for your comments. A few sentences have been added in “the section “Discussion”, along with 9 related references.
- All information from lines 399 to 405 is missing and MUST be added.
RESPONSE: Thank you for pointing this out. We have made the appropriate additions in the latest manuscript as you suggested.
- References: at least 23 of the 38 references are prior to the current decade (2020-2023). The authors must refresh this list with more updated references considering the amount of work published due to the importance of the subject. Moreover, some of the references are not complete (for example references 10 and 11).
RESPONSE: Thank you for pointing out this issue. We have fully checked and adjusted the references in our latest manuscript as you suggested.
Reviewer 3 Report
Comments and Suggestions for Authors
Result
Figure 1 was mentioned first time in Line 198. Please insert figure 1 after figure 1 was first time mentioned. Please describe figure 1 after figure 1 figure legend.
Same as all of figure 1-8
Figure 2 was mentioned first time in Line 217. Please insert figure 2 after figure 2 was first time mentioned.
Please describe figure 2 after figure 2 figure legend.
Figure 1,2,3,4,7,8
There are multi panels, please indicate them with a, b, c, d as template. Figure legend should be modified accordingly.
Figure 2-8
It is large figure, please use large figure legend starting from left side of marge 0 .
Author Response
Figure 1 was mentioned first time in Line 198. Please insert figure 1 after figure 1 was first time mentioned. Please describe figure 1 after figure 1 figure legend.
Same as all of figure 1-8
Figure 2 was mentioned first time in Line 217. Please insert figure 2 after figure 2 was first time mentioned.
Please describe figure 2 after figure 2 figure legend.
Figure 1,2,3,4,7,8
There are multi panels, please indicate them with a, b, c, d as template. Figure legend should be modified accordingly.
Figure 2-8
It is large figure, please use large figure legend starting from left side of marge 0 .
RESPONSE: Thank you for pointing out these problems. We have made changes to the placement and order of the figures, as well as the figure legends, in the latest manuscript as you suggested.
Round 2
Reviewer 3 Report
Comments and Suggestions for Authors
Authors did not changed the comments from previous review.
The second time review comments.
Melittin has strong Hemolysis.
authors mainly compare bioactivity of LC-AMP-F1 and Melittin.
Please include melittin in title, abstract, introduction
Please introduce Melittin
Introduction
Line 67- 70
It is great the aim of this study is to investigate the potential abilities of LC-AMP-F1 for antibacterial activity and cytotoxicity towards erythrocytes in vitro.
Please include objective such as 'antimicrobial resistance of LC-AMP-F1 and Melittin were analysed with bacterial strains from China General Microbiological Culture and clinic' to connect results after aim was introduced.
Method
please indicate what concentration was loaded in the analysis in the section 2.2
Line 171
'The samples were dried on silicon wafers, sprayed with gold, and observed on SEM'. Please write the detail of sample preparation
How the samples were dried? Where the samples on silicon wafer were located when it was dried? How long was taken to dry?
For spraying
How many amount of gold were sprayed on the surface of bacteria?
How long or what pressure were employed to spray? What is reference?
Which type of spray was employed? The SEM requested voltage and type of SEM from which company.
Line 174
The normal cells (LO2 and HEK293T) and cancer cells (4T1)
Please write culture condition and medium name
Result
Figure 1
Figure legend
'The hydrophobic residues are yellow, positively charged hydrophilic residues are blue, and negatively changed hydrophilic residues are red.'
There is no red color in the panel C. It is slight pink at letter Q. Please modify the panel C.
The number of y axis in panel D is very large.
Figure 7
Please make font size of scale bar bigger
Please recheck the size of E coli
Line 398
The reference 21 is not related with L. erythrognatha venom.
Reference 21 Huth HW, Santos DM, Gravina HD, Resende JM, Goes AM, De Lima ME, Ropert C: Upregulation of p38 pathway accelerates 479 proliferation and migration of MDA-MB-231 breast cancer cells. Oncology Reports 2017, 37(4):2497-2505.
Reference style
Please use MDPI reference style.
The following content is the comments from previous review.
Figure 1 was mentioned first time in Line 198. Please insert figure 1 after figure 1 was first time mentioned. Please describe figure 1 after figure 1 figure legend.
Same as all of figure 1-8
Figure 2 was mentioned first time in Line 217. Please insert figure 2 after figure 2 was first time mentioned.
Please describe figure 2 after figure 2 figure legend.
Figure 1,2,3,4,7,8
There are multi panels, please indicate them with a, b, c, d as template. Figure legend should be modified accordingly.
Figure 2-8
It is large figure, please use large figure legend starting from left side of marge 0 .
Author Response
Authors did not changed the comments from previous review.
The second time review comments.
- Melittin has strong Hemolysis.
authors mainly compare bioactivity of LC-AMP-F1 and Melittin.
Please include melittin in title, abstract, introduction
Please introduce Melittin
RESPONSE: We apologize that there was some misunderstanding during the first round of comment moderation and we got the wrong version uploaded. Thank you for your valuable comments again in this review. As your suggestion, we have added the introduction of Melittin in the abstract and introduction. But after much deliberation, we did not change the original title. The reason is that the main focus of this paper is the antimicrobial activity and anti-biofilm activity of LC-AMP-F1, and Melittin is only used as a positive control of the peptide. On the other hand, Melittin, as a bioactive peptide that has been extensively studied, has biological functions that include not only antibacterial, but also anti-tumor, anti-inflammatory, anti-viral, and other diverse biological activities, which are not discussed in this work. Therefore, we feel that the present work is not a complete and systematic comparative study and we would like to keep our original title. We would be grateful for your understanding.
- Introduction
Line 67- 70
It is great the aim of this study is to investigate the potential abilities of LC-AMP-F1 for antibacterial activity and cytotoxicity towards erythrocytes in vitro.
Please include objective such as 'antimicrobial resistance of LC-AMP-F1 and Melittin were analysed with bacterial strains from China General Microbiological Culture and clinic' to connect results after aim was introduced.
RESPONSE: We sincerely appreciate the valuable comments. We have already added the corresponding content in the introduction as “Hence, in this work, Melittin, a classical membrane-breaking AMP derived from honeybee venom, was set as a positive control [22-24]. And the antimicrobial resistance of LC-AMP-F1 and Melittin were analysed with bacterial strains from China General Microbiological Culture and clinic.” in the latest version of manuscript.
- Method
please indicate what concentration was loaded in the analysis in the section 2.2
RESPONSE: Thank you for the reminder. The concentration of peptide was added correspondingly in the latest version of manuscript.
- Line 171
'The samples were dried on silicon wafers, sprayed with gold, and observed on SEM'. Please write the detail of sample preparation
How the samples were dried? Where the samples on silicon wafer were located when it was dried? How long was taken to dry?
For spraying
How many amount of gold were sprayed on the surface of bacteria?
How long or what pressure were employed to spray? What is reference?
Which type of spray was employed? The SEM requested voltage and type of SEM from which company.
RESPONSE: We sincerely thank the reviewer for careful reading. Details of the operation of the electron microscope experiments we add in the appropriate place as “The samples were transfer and dried on silicon wafers at 60℃, for 5min. Then Specimens are attached to metallic stubs using carbon stickers and sputter-coated with gold for 30s by Lon Sputtering Apparatus (HITACH MC1000). At last, the samples were observed by SEM (HITACHI Regulus 8100, 1 kV) [27,28].”.
- Line 174
The normal cells (LO2 and HEK293T) and cancer cells (4T1)
Please write culture condition and medium name
RESPONSE: Thanks for your careful checks. The culture condition and medium types have added in the section 2.11.
- Result
Figure 1
Figure legend
'The hydrophobic residues are yellow, positively charged hydrophilic residues are blue, and negatively changed hydrophilic residues are red.'
There is no red color in the panel C. It is slight pink at letter Q. Please modify the panel C.
The number of y axis in panel D is very large.
RESPONSE: We feel sorry for our carelessness. In fact, through the online software analysis, it shows pink, which is not properly expressed here. It has been changed to pink in Figure 1. We have reduced the horizontal and vertical fonts in Figure 1d.
- Figure 7
Please make font size of scale bar bigger
Please recheck the size of E coli
RESPONSE: Thank you for your suggestion. We have checked the size of E coli and doubled the size of the scale.
- Line 398
The reference 21 is not related with L. erythrognatha venom.
Reference 21 Huth HW, Santos DM, Gravina HD, Resende JM, Goes AM, De Lima ME, Ropert C: Upregulation of p38 pathway accelerates 479 proliferation and migration of MDA-MB-231 breast cancer cells. Oncology Reports 2017, 37(4):2497-2505.
RESPONSE: Thanks for your careful checks. The reference 21 is related with LyeTx II from L. erythrognatha venom as mentioned in the introduction. In addition, we have added the reference 20.
- Reference style
Please use MDPI reference style.
RESPONSE: Thanks for pointing that out. The reference style were adjusted in MDPI style.
The following content is the comments from previous review.
- Figure 1 was mentioned first time in Line 198. Please insert figure 1 after figure 1 was first time mentioned. Please describe figure 1 after figure 1 figure legend.
Same as all of figure 1-8
RESPONSE:Thanks for pointing that out. We have corrected it in the latest version.
- Figure 2 was mentioned first time in Line 217. Please insert figure 2 after figure 2 was first time mentioned.
Please describe figure 2 after figure 2 figure legend.
RESPONSE:Thanks for pointing that out. We have corrected it in the latest version.
- Figure 1,2,3,4,7,8
There are multi panels, please indicate them with a, b, c, d as template. Figure legend should be modified accordingly.
RESPONSE:Thanks for pointing that out. We have corrected it in the latest version.
- Figure 2-8
It is large figure, please use large figure legend starting from left side of marge 0 .
RESPONSE:Thanks for pointing that out. We have corrected it in the latest version.
Round 3
Reviewer 3 Report
Comments and Suggestions for Authors
Authors have improved manuscript.
There is miss understanding regarding the figure style. In the manuscript v1, There were large panel figure. I suggested figure legend starting from left margin. You have improve figures in v2.
In the current version, all of the figure have been moved to the left margin. I would suggest please move all of figure left side starting from left margin 4.6 cm.